# Bright Single-Photon Emitters with a CdSe Quantum Dot and Multimode Tapered Nanoantenna for the Visible Spectral Range

**DOI:** 10.3390/nano11040916

**Published:** 2021-04-03

**Authors:** Maxim Rakhlin, Sergey Sorokin, Dmitrii Kazanov, Irina Sedova, Tatiana Shubina, Sergey Ivanov, Vladimir Mikhailovskii, Alexey Toropov

**Affiliations:** 1Ioffe Institute, 194021 St. Petersburg, Russia; sorokin.beam@mail.ioffe.ru (S.S.); kazanovdr@gmail.com (D.K.); irina@beam.ioffe.ru (I.S.); shubina@beam.ioffe.ru (T.S.); ivan@beam.ioffe.ru (S.I.); toropov@beam.ioffe.ru (A.T.); 2Faculty of Physics, St. Petersburg State University, 199034 St. Petersburg, Russia; zihertge@gmail.com

**Keywords:** quantum dots, single-photon emission, CdSe, MBE, nanoantenna

## Abstract

We report on single photon emitters for the green-yellow spectral range, which comprise a CdSe/ZnSe quantum dot placed inside a semiconductor tapered nanocolumn acting as a multimode nanoantenna. Despite the presence of many optical modes inside, such a nanoantenna is able to collect the quantum dot radiation and ensure its effective output. We demonstrate periodic arrays of such emitters, which are fabricated by focused ion beam etching from a II-VI/III-V heterostructure grown using molecular beam epitaxy. With non-resonant optical pumping, the average count rate of emitted single photons exceeds 5 MHz with the second-order correlation function $g^{\left(2\right)}$(0) = 0.25 at 220 K. Such single photon emitters are promising for secure free space optical communication lines.

## 1. Introduction

Quantum information and communication systems require non-classical light sources that can emit, upon request, either one photon or a pair of entangled single photons with high external quantum efficiency at a certain frequency [1,2]. Self-organizing single quantum dots (QDs) grown by various epitaxial methods are promising candidates for creating single-photon emitters (SPE) due to the small emission line width, fast radiation decay time, high and stable quantum efficiency, and the ability to be integrated with semiconductor electronics devices [3]. In principle, single-photon emission can be obtained in a wide spectral range from medium ultraviolet to the optical telecommunication C-band (1.55 µm) through optical pumping of single QDs fabricated in different material systems [4,5,6,7,8]. Most of such emitters operate at cryogenic temperatures and only the epitaxial QDs made of wide-band-gap materials can work as SPE up to room temperature [9,10]. Alternative systems for producing single-photon emission at elevated temperatures could be single molecules [11], diamond color centers [12], and colloidal quantum dots [13]. However, the radiative lifetimes of these emitters are longer than several nanoseconds, which prevents the achievement of a sufficiently high generation rate and purity of single-photon emission [14]. Furthermore, colloidal QDs are subject to blinking—this problem has not been completely resolved up to now [15]. In addition, for most of the above-mentioned systems, at present, there is no reproducible technology allowing fabrication of relevant photonic structures of which the quality is sufficient for obtaining SPE with required purity and brightness of single-photon emission. In fact, the only commercially available triggered SPE takes advantage of a monolithic optical microcavity with an embedded single InAs/GaAs QD to generate single photons in a very narrow spectral range around 920 nm at low temperatures (<8 K) [16]. On the other hand, of particular interest for many applications is the green-red wavelength range (500–700 nm), which corresponds to the region of the greatest sensitivity of modern Si-based single-photon avalanche photodiodes and is also suitable for the development of protected free-space optical communication lines [1].

It should be noted that the high-temperature generation of single photons relies on precise spectral separation of narrow emission lines originating from excitons and different electron-hole complexes confined in a single QD, such as biexcitons and trions [17]. With increasing temperature, the emission lines spectrally widen and at a certain temperature they start to overlap, which makes it difficult to obtain pure single-photon statistics. QDs based on II-VI wide-gap compounds are considered promising candidates for creating single-photon sources operating at room temperature due to the especially high biexciton and exciton binding energies and, as a consequence, the particularly large spectral distance between the corresponding emission lines [17,18,19]. To date, the room temperature green single-photon emission has been already obtained both in a structure with CdSe QDs formed in a ZnSe nanowire [20] and in an epitaxial heterostructure with CdSe/ZnSe QDs, where emission of a single QD was spatially separated due to the use of a submicron aperture opened in the non-transparent metal mask [10].

The single-photon generation rate in these structures, however, did not exceed tens of kilohertz, which is two-three orders of magnitude lower than the speed required for applications in quantum cryptography systems. This parameter depends mainly on the efficiency of the radiation collection and output. For conventional epitaxial InAs/GaAs QDs, the improved collection of the single-QD emission is usually achieved by fabricating columnar micropillars with distributed GaAs/AlAs Bragg mirrors consisting of the layers with close lattice constants [21]. The fabrication of such structures in the systems based on II-VI semiconductors is complicated by the absence of corresponding binary lattice-matched compounds. Lohmeyer et al. managed to fabricate an epitaxial microcavity structure with distributed Bragg mirrors consisting of alternately strained MgS and ZnCdSe layers, which included CdSe/ZnSSe QDs as an active region [22]. Unfortunately, the MgS compound is rapidly oxidized in air, which greatly hinders the subsequent processing of the grown heterostructures and, as a result, leads to the rapid degradation of manufactured devices.

A promising alternative strategy in achieving the effective light extraction for II–VI QDs can rely on fabrication of dielectric photonic nanoantennas. These nanostructures are columnar (cylindrical) waveguides with variable cross-section. The base (maximum) diameter of the column is low enough to ensure support only of a fundamental guided mode (HE${}_{11}$), providing its strong confinement and coupling to the QD emission. The mode propagation along the smoothly tapered cylindrical waveguide ensures its adiabatic conversion into a strongly deconfined mode with a narrower, Gaussian-like far-field distribution. This approach allows very high photon extraction efficiency (>90%) and their efficient collection by standard optics in a relatively wide spectral range of several tens of nanometers [23]. Claudon et al. managed to fabricate photonic nanoantennas in the form of a nanowire with a diameter of about 200 nm that is acceptable for the emission wavelength of InAs/GaAs QDs [24]. To collect the maximum amount of light, a photonic nanowire was installed on a gold-covered, reflecting substrate. For CdSe QDs emitting in the wavelength range of 500–550 nm, the optimal nanoantenna diameter should be less than 130 nm, which complicates the fabrication technology. In addition, the formation of a metal mirror on the back side of a nanowire requires development of a laborious technology that is not suitable for mass production. Nevertheless, we partly implemented this approach previously, fabricating a hybrid semiconductor/dielectric 200 nm thick photonic nanoantenna with an embedded CdSe/ZnSe QD and without any backside mirror, which allowed us to obtain the single-photon emission with a count rate of about 1 MHz at 80 K [25].

In this paper, we design and fabricate a SPE based on a CdSe/ZnSe QD placed within a purely semiconductor photonic nanoantenna fabricated by focused ion-beam (FIB) etching from a III–V/II–VI heterostructure grown by molecular beam epitaxy (MBE). The diameter of the nanoantenna was large enough to support multiple optical modes. The available technology did not allow us to reproduce the metal mirror on the back side of the nanowire. By analyzing numerically the confinement and reflection of light inside, as well as its output to the far optical field, we, nevertheless, demonstrated that the characteristics of such a nanoantenna are sufficient to implement an efficient single-photon source in the green-yellow visible range. The experimental realization of the SPE with an average radiation rate exceeding 5 MHz at a practically important temperature of 220 K confirms the fruitfulness of this approach.

## 2. Single Photon Emitters with Multimode Nanoantenna

A purely semiconductor SPE developed in this work is schematically shown in Figure 1a. It represents a truncated cone-shaped nanocolumn with the bottom diameter d∼400 nm, upper diameter of about 50 nm, and a height of 1600 nm. These sizes correspond to experimental values, which are strictly limited by capabilities of the employed FIB etching technique. The emitting QD is located inside the II-VI (ZnMgSSe) conical part, 100 nm above the GaAs buffer layer (the conical part consisting of GaAs is 200 nm high). For a rough estimation of the mode structure, we approximated a cone-shaped nanocolumn by a cylindrical waveguide. We performed calculations of the dispersion curves for the lowest waveguide modes using a finite-difference mode solver in Comsol Multiphysics. The theoretical description of the waveguide modes was similar to that published in [26,27]. The refractive index of the material of the cylinder was taken as 2.7. In calculations, the cylinder was located in the air surrounded by a “perfectly matched layer” (PML). The cylinder radius was varied from 65 to 220 nm. Calculations show that the basal part of the nanocolumn with the radius of about 200 nm supports not only the fundamental HE${}_{11}$ mode, but many (up to ten) modes of higher orders. The cylinder thinner than 75 nm supports only a single fundamental mode. The calculated dispersion curves for the three lowest-order modes are shown in Figure 1b. In the narrowing upper part of the nanocolumn, the only supported mode is the HE${}_{11}$ one. It efficiently leaks outside, transforming into a Gaussian-like plane wave front, like described in [28].

To find the distribution of the electromagnetic field produced at a fixed frequency by a QD (an oscillating dipole) located at a specific height along the central axis of a nanocolumn, we carried out a 3D finite difference frequency domain simulation in Comsol multiphysics for calculating both near- and far-field radiation intensities. We used the in-plane point-dipole source perpendicular to the cylinder axis to model the spontaneous emission of the exciton optical transition. The frequency dispersion of the refractive index of GaAs was taken into account. The volume taken for the modeling was a cylindrical box with a 3000 nm height and 700 nm radius. The system is surrounded by the PML. We used a physics-controlled mesh that depends on a wavelength and has minimum element sizes of ∼1 nm. The far-field calculations were done by the built-in Stratton–Chu formula that allows one to make a transformation of the near field into the far field. Such approach allowed us to simulate optical properties of the studied structures with specific geometries and reach the best results for specific goals similar to how it was done in [29,30,31].

An example of near-field electromagnetic distribution is depicted in Figure 2a. The calculated field distribution clearly displays interference features indicating efficient reflection of the involved waveguide modes from both the top and bottom of the nanocolumn. The role of the bottom “mirror” is played mainly by a sharp change in both real and imaginary parts of the refractive index at the interface between II-VI and III-V materials. To a certain extent, this circumstance compensates the absence of a metal mirror at the bottom of the nanocolumn.

Figure 2b shows a normalized modulus of the electric field inside a 1.6 µm-long nanocolumn, calculated for several wavelengths, dependent on the CdSe QD position. To perform this calculation, we used an approach based on the Lorentz reciprocity theorem, as in [32,33]. We solved the reverse scattering problem: the plane wave of a fixed frequency is scattered by a nanocolumn with a QD inserted on the main axis of the nanocolumn at different distances from the nanocolumn bottom. For each wavelength, we observe interference fringes defined by the sum of the spatial distribution of all supported optical modes inside the nanocolumn. A decrease in the strength of the electromagnetic field at the QD positions higher than 500 nm is qualitatively associated with the narrowing of the nanocolumn towards its apex, which leads to sequential cutting off the higher-order modes demonstrating stronger reflection from the nanorod bottom.

While the comprehensive analysis of the individual contributions of the involved waveguide modes to the light collection and extraction is out of the scope of this work, the presented results of the numerical calculation provide useful hints about both the optimal position of the emitting QD inside the nanocolumn and relative tolerance of this parameter to the emission wavelength. In particular, it is clear from Figure 2b that the optimal QD location is at the height between 300 and 550 nm for a wide spectral range between 490 and 550 nm. One should note that the radiation wavelength of the CdSe/ZnSe QDs can be continuously tuned from 460 to 540 nm by changing the nominal thickness of CdSe inserts during the MBE growth. The respective calibration curve obtained as a result of many technological experiments is presented in Figure 2c. Thus, by varying the nanocolumn diameter and height, the QD position inside, and growth parameters during the QD formation, it is possible, in principle, to obtain the nearly optimal set of parameters providing efficient generation of single-photon emission in the whole green and most of the yellow parts of the visible spectrum. The geometrical parameters of the fabricated structures were taken as a compromise between the results of the theoretical optimization and limitations imposed by the employed fabrication technologies, first of all, by the FIB etching.

For the chosen set of parameters, we compared the calculated far-field radiation of the nanocolumn, collected by a typical objective lens with numerical aperture NA = 0.42, and a planar structure with the same oscillating dipole. It turns out to be 27 times more intensive for the nanocolumn due to the efficient collection of the QD emission into the guided modes and better directivity of the output radiation.

## 3. Sample Fabrication

The investigated heterostructure was grown using MBE technology on a GaAs (001) substrate with a 200-nm-thick buffer layer of GaAs [34]. A 10-nm-thick ZnSe layer, a short-period ZnSe (1.8 nm)/ZnSSe (1.5 nm) superlattice with a total thickness of 33 nm, and a 50-nm-thick ZnMgSSe layer (lattice-matched to GaAs, E${}_{g}$∼3.0 eV) were grown on top of the GaAs buffer. Next, CdSe QDs were grown between wide-gap barriers consisting of 1.5-nm-thick ZnSSe and 5-nm-thick ZnSe layers. The structure growth was completed with the deposition of the layers of 1.3-µm-thick ZnMgSSe and 3-nm-thick ZnSe (Figure 3a).

As previously demonstrated, this design provides large enough band offsets at the QD interfaces that prevents thermal escape of charge carriers from the QD into the surrounding barriers up to room temperature [25]. The CdSe QDs were formed in a thermal activation mode, which implies variation of the surface energy, facilitating the 2D–3D transition of the strained CdSe layer [35,36,37]. This technique allows one to obtain a sufficiently low QDs density (∼10${}^{10}$ cm${}^{-2}$) that complies with spatial isolation within the 400-nm-thick nanocolumn of only a few single QDs. The contrast analysis of high-resolution transmission electron microscopy (HRTEM) images has revealed that the active QD layer consists of Cd-rich regions with a lateral size of 10–15 nm and a height of ∼3 nm, embedded in a 7–10 mL thick graded CdZnSe quantum well with a lower content of Cd (Figure 3b). These results are in good agreement with the HRTEM data reported earlier for the CdSe/ZnSe QDs grown in a similar manner [38,39].

The photonic nanoantennas were fabricated from this heterostructure by using two sequential passes of the focused Ga-ion beam etching with currents 360 and 8 pA, respectively. First, a crater with a lateral size of 10 µm and a cone in the center were manufactured. Then, after reducing the current, the photonic nanoantenna was etched down precisely to the required sizes. As a result, an array of cone-shape structures with a varying cross-section, a base diameter of 400 nm, and a height of about 1.6 µm were obtained (Figure 3c,d).

We should note that the thermal-activation method does not allow one to control precisely the emission wavelength of each fabricated QD, as well as its spatial location. As a result, the QDs are randomly located within the growth plane and the emission spectrum averaged over a large number of QDs is represented by a wide (∼300 meV) spectral band. Currently, we do not adjust the nanoantenna position with respect to the position of a particular QD. Therefore, fabrication of an optimized single-photon source relied on the manufacture of many devices and further search of the structure with satisfactory characteristics. The natural extension of this technique implies the implementation of cryolithography, as it was demonstrated for post microcavities with InAs/GaAs QDs [40].

## 4. Measurements and Quantum Statistics

A peculiarity of FIB etching is the emergence in the vicinity of the etched region of a high density of point defects working as centers of nonradiative recombination. One possible solution to this problem is the annealing of the non-equilibrium defects by laser radiation [41]. The structure exposure to high enough cw laser (λ = 405 nm) power density (above 50 W/cm${}^{2}$) caused a noticeable (a few times) increase in the PL intensity of single QDs during the first few seconds that followed the signal saturation. After annealing, the PL signal remained stable at any reasonable excitation powers.

For micro-photoluminescence (µ-PL) measurements, the samples were mounted in a He-flow cryostat ST-500-Attocube (Janis) with a XYZ piezodriver inside, which allowed us to optimize and precisely maintain the positioning of a chosen place on the sample with respect to a laser spot. The luminescence was dispersed by a 0.5 m monochromator with a 1200/mm grating, providing direct spectral resolution ∼70 µeV. Non-resonant optical excitation by a cw laser line (405 nm) was used with the laser power density below 4 W/cm${}^{2}$. A typical µ-PL spectrum of a single photonic structure includes narrow lines related to the emission of several individual QDs located inside the nanoantenna. The full width at half maximum of an emission line increases from 0.4 meV at 10 K up to 9 meV at 220 K. To isolate the chosen excitonic line from emission of other QDs we used long-pass and short-pass tunable interference optical filters (Figure 4a,b). To investigate the photon correlations statistics we used a Hanbury Brown-Twiss setup with two single-photon avalanche photodiodes (SPAD) and a 50:50 beam splitter (Figure 4c). The time intervals between the detection events are registered by an electronic start-and-stop scheme (SPC-130, Becker and Hickle). The resultant histogram of coincidences, plotted as a function of the delay time between the detection events, is proportional to the second-order correlation function $g^{\left(2\right)}\left(\tau \right)$ [42,43]. To determine $g^{\left(2\right)}\left(\tau \right)$ on an absolute scale, it is necessary to normalize the measured coincidence statistics and take into account correlations associated with the presence of a background signal. The function is normalized according to the formula:(1)$$\begin{array}{c} C_{N}\left(t\right)=c\left(t\right)/\left(N{}_{1}N_{2}T\omega \right), \end{array}$$
where c(t) is raw coincidence number, $N_{1,2}$ are single counters rates, ω is the width of a time bin, and *T* is a total acquisition time.

Figure 4d represents the $g^{\left(2\right)}\left(\tau \right)$ function measured for a chosen excitonic line at 220 K. The function is determined as the normalized statistics of coincidences after taking into account background correlations as
(2)$$\begin{array}{c} g^{\left(2\right)}\left(\tau \right)=\left[C_{N}\left(t\right)-\left(1-\rho^{2}\right)\right]/\rho^{2}, \end{array}$$
(3)$$\begin{array}{c} \rho =S/(S+B), \end{array}$$
where *S* is a signal from QD and *B* is a background level. ρ is measured independently in each experimental run [43]. The measured second order correlation function can be approximated using the equation:(4)$$\begin{array}{c} g^{\left(2\right)}\left(\tau \right)=a-be^{\left(-\tau /c\right)}, \end{array}$$
where *a*, *b* and *c* are fitting parameters. The value c=400 ps gives an estimate for the dip width of the correlation function at zero delay, which directly correlates with the radiative lifetime of the exciton. Such obtained value of the lifetime is consistent with the data reported previously [44]. The approximation of the experimental data using Equation (Equation 4) gave the value of the correlation function $g^{\left(2\right)}\left(0\right)=0.25\pm 0.04$, which is clear evidence of the single-photon nature of the emission [45]. Above 220 K, the background of the µ-PL spectrum becomes more pronounced due to mainly emergence of acoustic-phonon sidebands of the neighboring emission lines, causing the appearance of uncorrelated photons and, as a consequence, a decreased purity of single-photon emission [19].

The average number of photons recorded in this structure by one detector per second was 2.5 × 10${}^{5}$ both for 10 and 220 K, which, taking into account the hardware function of the measuring equipment, corresponds to the intensity of single-photon emission on the first lens as large as 5 MHz. Close values of the emission rate measured at different temperatures indicate a relatively small number of active defects, as well as adequacy of the implemented design. The application of a semiconductor photonic nanoantenna made it possible to increase the emission intensity by 10 times in comparison with the previously investigated mesa-structure [46] and by 5 times in comparison with a semiconductor/dielectric nanoantenna [25].

## 5. Conclusions

We have demonstrated successful operation in the visible range of single photon sources with CdSe QDs and purely semiconductor multimode optical nanoantennas. The columnar sources were fabricated from an MBE-grown heterostructure using FIB etching. A decrease in the QDs density made it possible to study single emission lines of individual QDs. Despite the fact that these nanoantennas support several optical modes, they are able to collect and directionally extract light, providing the radiation intensity in the far field ten times higher than from a mesa-structure with similar QDs. The realized SPEs have demonstrated single-photon emission with an average count rate more than 5 MHz at a practically important temperature of 220 K, which can be achieved with a conventional thermoelectric cooler. The value of second-order correlation function $g^{\left(2\right)}\left(0\right)$ for the best investigated structure was about 0.25. A further increase in the purity of the measured $g^{\left(2\right)}$ function can be achieved by using resonant optical pumping [47], while creating a single-photon source with a higher emission intensity requires even better control of the photonic nanoantenna shape during post-growth processing. The application of advanced cryolithography would allow precise alignment of the spatial positions of the QD and nanocolumn in the plane. We believe that expected progress in nanotechnology will make possible the SPE applications in quantum cryptography systems needed for free space optical communication lines.

## Figures and Tables

**Figure 1 nanomaterials-11-00916-f001:**
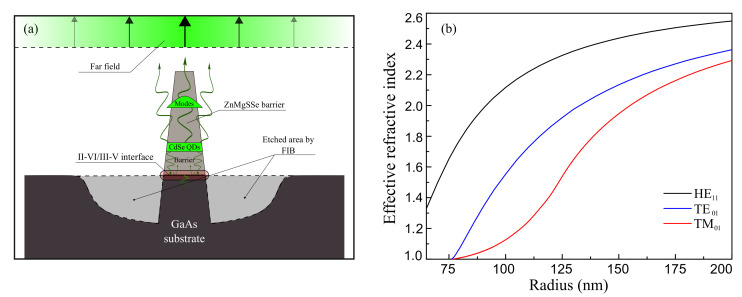
(**a**) Schematic of a photonic nanoantenna with inserted quantum dots (QDs). (**b**) Dispersion curves of 3 lowest modes in a cylindrical waveguide, calculated versus radius for a typical wavelength of 520 nm.

**Figure 2 nanomaterials-11-00916-f002:**
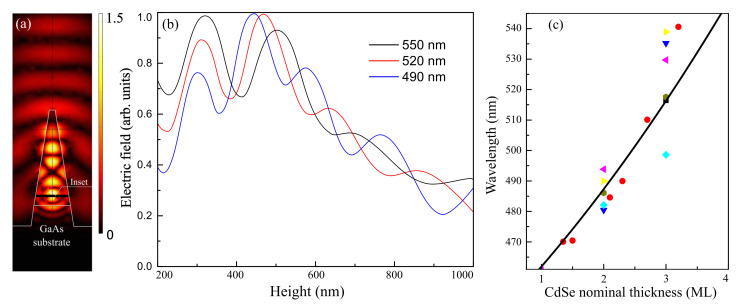
Typical electromagnetic field distribution inside and outside of the nanocolumn at the wavelength of 520 nm. (**a**) Color bar shows the normalized values of modulus of the electric field. (**b**) The modulus of the electric field dependence on the CdSe QD height inside a 1.6 µm-long nanocolumn, calculated for wavelengths marked in a plot and normalized to the maximum of the amplitude for each wavelength. (**c**) Dependence of the radiation wavelength of CdSe/ZnSe QDs on the nominal thickness of a CdSe layer, deposited at the molecular beam epitaxy (MBE) growth, given in monolayers. Different symbols mark various growth runs. The PL measurements were performed at 77 K.

**Figure 3 nanomaterials-11-00916-f003:**
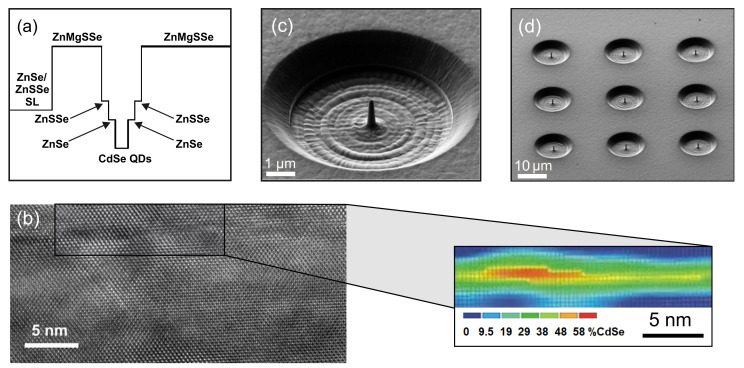
(**a**) Energy-band diagram of a CdSe QD confined between wide-gap barriers. (**b**) Cross-section high-resolution transmission electron microscopy (HRTEM) image of a heterostructure with CdSe QDs (darker spots). Inset shows Cd distribution obtained from the analysis of the contrast of a HRTEM image. (**c**,**d**) Scanning electron microscopy images of typical devices recorded at different magnifications.

**Figure 4 nanomaterials-11-00916-f004:**
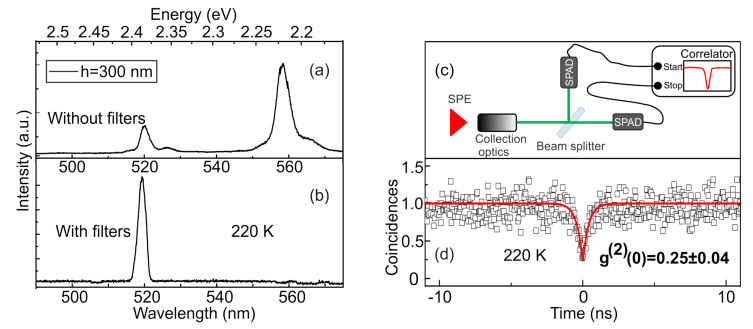
(**a**,**b**) Representative µ–PL spectra of a photonic nanoantenna with CdSe QDs, measured at 220 K without (**a**) and with (**b**) tunable long-pass and short-pass optical filters. (**c**) Schematic representation of the Hanbury Brown and Twiss setup used for measuring the second-order correlation function $g^{\left(2\right)}$. (**d**) Normalized second-order correlation function $g^{\left(2\right)}$ of single photon emission, measured at 220 K for a spectrally filtered single excitonic line. The obtained value of $g^{\left(2\right)}$(0) is 0.25.

## Data Availability

The data that supports the findings of this study are available within the article.

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
