# Peer review of "Bright Single-Photon Emitters with a CdSe Quantum Dot and Multimode Tapered Nanoantenna for the Visible Spectral Range"

_nanomaterials, 2021, doi:10.3390/nano11040916_

Round 1
Reviewer 1 Report
Rakhlin and co-authors report a characterization of the photon source consisting of a quantum dot embedded within an antenna, that enables the far-field coupling of light emitted by the QD itself. The approach of using antennas to collect radiation emitted by nanocrystals is well know, however the work is an interesting one, and the results will be of interst for the scientific community working in the fields of nanophotonics and nanomaterials.
The scholarly presentation of the results is good, the topic is well introduced and the results are discussed in a clear way. I recommend this manuscript for publication, pending minor editing of the English.
Author Response
Reviewer 1
Rakhlin and co-authors report a characterization of the photon source consisting of a quantum dot embedded within an antenna, that enables the far-field coupling of light emitted by the QD itself. The approach of using antennas to collect radiation emitted by nanocrystals is well know, however the work is an interesting one, and the results will be of interst for the scientific community working in the fields of nanophotonics and nanomaterials.
The scholarly presentation of the results is good, the topic is well introduced and the results are discussed in a clear way. I recommend this manuscript for publication, pending minor editing of the English.
Answer:
Thank you very much for reading our manuscript carefully and for your appreciation of our work. In accordance with the recommendation, the English has been polished and several typos have been corrected.
Reviewer 2 Report
In this paper, the authors design and fabricate a SPE based on a CdSe/ZnSe QD placed within a purely semiconductor photonic nanoantenna fabricated by focused ion-beam (FIB) etching from a III-V/II-VI heterostructure grown by molecular beam epitaxy (MBE).The diameter of the nanoantenna was large enough to support multiple optical modes. By analyzing numerically the confinement and reflection of light inside, as well as its output to the far optical field, it is demonstrated that the characteristics of such a nanoantenna are sufficient to implement an efficient single-photon source in the green-yellow visible range. The experimental realization of the SPE with an average radiation rate exceeding 5 MHz at a practically important temperature of 220 K confirms the fruitfulness of this approach. The authors have done a great job to numerical analysis, fabricate, and measure the devices. If the authors can solve the following concerns, it is highly recommend to accept the manuscript for nanomaterials after minor revision.
- Some English grammars are inaccurate and need to be modified.
- There are too few points in the curve of Fig. 2B, so the author needs to re simulate it.
- The characterization of CdSe QDs is not detailed enough, and the author needs to provide high-resolution mapping.
- The specific parameters of the simulation are not detailed enough, and the author should refer to the relevant simulation literature, such as: Physica E: Low-dimensional Systems and Nanostructures, 127 (2021) 114526; Diamond & Related Materials 111 (2021) 108227; Chinese Physics B 2021, 30 (2), 024211.
Author Response
Reviewer 2
In this paper, the authors design and fabricate a SPE based on a CdSe/ZnSe QD placed within a purely semiconductor photonic nanoantenna fabricated by focused ion-beam (FIB) etching from a III-V/II-VI heterostructure grown by molecular beam epitaxy (MBE).The diameter of the nanoantenna was large enough to support multiple optical modes. By analyzing numerically the confinement and reflection of light inside, as well as its output to the far optical field, it is demonstrated that the characteristics of such a nanoantenna are sufficient to implement an efficient single-photon source in the green-yellow visible range. The experimental realization of the SPE with an average radiation rate exceeding 5 MHz at a practically important temperature of 220 K confirms the fruitfulness of this approach. The authors have done a great job to numerical analysis, fabricate, and measure the devices. If the authors can solve the following concerns, it is highly recommend to accept the manuscript for nanomaterials after minor revision.
- Some English grammars are inaccurate and need to be modified.
- There are too few points in the curve of Fig. 2B, so the author needs to re simulate it.
- The characterization of CdSe QDs is not detailed enough, and the author needs to provide high-resolution mapping.
- The specific parameters of the simulation are not detailed enough, and the author should refer to the relevant simulation literature, such as: Physica E: Low-dimensional Systems and Nanostructures, 127 (2021) 114526; Diamond & Related Materials 111 (2021) 108227; Chinese Physics B 2021, 30 (2), 024211.
Answer:
Thank you very much for careful reading of our manuscript. We highly appreciated all comments and recommendations, which helped us to significantly improve the paper. We submit a revised version of the manuscript, a point-by-point response, and the manuscript version where all the changes are marked with red color.
Q1. Some English grammars are inaccurate and need to be modified.
A1. The English has been polished and several typos have been corrected.
Q2. There are too few points in the curve of Fig. 2B, so the author needs to re simulate it.
A2. We have recalculated our curves with higher accuracy. New results are represented in Fig. 2 (b).
Q3. The characterization of CdSe QDs is not detailed enough, and the author needs to provide high-resolution mapping.
A3. HRTEM image and contrast analysis are represented in Fig. 3. (b). The comment is added in the Section “Sample fabrication” of the revised manuscript. To strengthen our consideration, we introduced two additional references [38],[39].
Q4. The specific parameters of the simulation are not detailed enough, and the author should refer to the relevant simulation literature, such as: Physica E: Low-dimensional Systems and Nanostructures, 127 (2021) 114526; Diamond & Related Materials 111 (2021) 108227; Chinese Physics B 2021, 30 (2), 024211.
A4. Additional parameters of the simulation are added in the “Single photon emitters with multimode nanoantenna” Section of the revised manuscript. Also, additional references are added (see References [29-31]).
Reviewer 3 Report
The authors reported single photon emitters with tapered nanoantenna. The work is interesting with some novelty, and could be publishable after addressing the following issues. 1. The authors discussed the importance of nanoantenna. However, no control experiments are shown to study the effects of the nanoantenna. It is suggested to compare single photon emitters without the nanoantenna, or with a different design. This will be helpful to evaluate the importance of the nanoantenna. 2. More experimental details should be provided. For example, the authors claimed that "the second-order correlation function g(2)(t) obtained for this excitonic line at 220 K using a standard Hanbury Brown and Twiss set up". More details about the "standard" Hanbury Brown and Twiss set up should be provided.Author Response
Reviewer 3
The authors reported single photon emitters with tapered nanoantenna. The work is interesting with some novelty, and could be publishable after addressing the following issues. 1. The authors discussed the importance of nanoantenna. However, no control experiments are shown to study the effects of the nanoantenna. It is suggested to compare single photon emitters without the nanoantenna, or with a different design. This will be helpful to evaluate the importance of the nanoantenna. 2. More experimental details should be provided. For example, the authors claimed that "the second-order correlation function g(2)(t) obtained for this excitonic line at 220 K using a standard Hanbury Brown and Twiss set up". More details about the "standard" Hanbury Brown and Twiss set up should be provided.
Answer:
Thank you very much for careful reading of our manuscript. We highly appreciated extremely helpful comments and recommendations. The needed corrections are done in the revised version of the manuscript (marked with red color). Please, find below a point-by-point response to the particular comments.
Q1. The authors discussed the importance of nanoantenna. However, no control experiments are shown to study the effects of the nanoantenna. It is suggested to compare single photon emitters without the nanoantenna, or with a different design. This will be helpful to evaluate the importance of the nanoantenna.
A1. We have compared our semiconductor photonic nanoantenna with a mesa-structure and a semiconductor/dielectric photonic nanoantenna, which we fabricated and studied previously. The applying of a semiconductor photonic nanoantenna made it possible to increase the emission intensity by 10 times in comparison with the previously investigated mesa-structure [Rakhlin et al., J. Phys.: Conf. Ser. 2018, 993, 012023] and by 5 times in comparison with a semiconductor/dielectric nanoantenna [Rakhlin et al., JETP Letters 2018, 108, 201–205]. This comment is provided in the Section “Measurements and quantum statistics”.
Q2. More experimental details should be provided. For example, the authors claimed that "the second-order correlation function g(2)(t) obtained for this excitonic line at 220 K using a standard Hanbury Brown and Twiss set up". More details about the "standard" Hanbury Brown and Twiss set up should be provided.
A2. More experimental details on measurement techniques are provided in the Section 4 “Measurements and quantum statistics”. In particular, the scheme of Hanbury Brown and Twiss setup is represented now in Fig. 4c and discussed in the text.
Round 2
Reviewer 3 Report
The authors have revised the manuscript properly and warrant publication now.